# Learning Deep Bilinear Transformation for Fine-grained Image Representation

**Heliang Zheng**[1]*, **Jianlong Fu**[2], **Zheng-Jun Zha**[1], **Jiebo Luo**[3]
[1]University of Science and Technology of China, Hefei, China
[2]Microsoft Research, Beijing, China
[3]University of Rochester, Rochester, NY
[1]zhenghl@mail.ustc.edu.cn, [2]jianf@microsoft.com, [1]zhazj@ustc.edu.cn, [3]jluo@cs.rochester.edu

## Abstract

Bilinear feature transformation has shown the state-of-the-art performance in learning fine-grained image representations. However, the computational cost to learn pairwise interactions between deep feature channels is prohibitively expensive, which restricts this powerful transformation to be used in deep neural networks. In this paper, we propose a deep bilinear transformation (DBT) block, which can be deeply stacked in convolutional neural networks to learn fine-grained image representations. The DBT block can uniformly divide input channels into several semantic groups. As bilinear transformation can be represented by calculating pairwise interactions within each group, the computational cost can be heavily relieved. The output of each block is further obtained by aggregating intra-group bilinear features, with residuals from the entire input features. We found that the proposed network achieves new state-of-the-art in several fine-grained image recognition benchmarks, including CUB-Bird, Stanford-Car, and FGVC-Aircraft.

## 1 Introduction

Fine-grained image recognition aims to distinguish subtle visual differences within a subcategory (e.g., various bird species [1, 2], and car models [3, 4]). Predominant approaches are often divided into two streams, as the following distinct characteristics existed in fine-grained recognition datasets: 1) well-structured, e.g., different birds share similar semantic parts like head, wings and tail [5–9]), 2) rich details, e.g., sophisticated textures are supposed to be useful to distinguish two similar species [10–13]. Part-based approaches were first proposed to focus on semantic attention, which particularly decompose a fine-grained object into significant parts, and train several subsequent discriminative part-nets for classification. Such architectures usually result in suboptimal performance, as the power of end-to-end optimization has not been fully studied.

The state-of-the-art results have been achieved by bilinear feature transformation, which learns fine-grained details over a global image by calculating pairwise interactions between feature channels in fully-connected layers. However, the exponential growth over feature dimensions (e.g., $N$ times increase for input channels of $N$) restricts this powerful transformation to be used in deep neural networks. To solve the high-dimensionality issue, compact bilinear [14] and low-rank bilinear [15, 16] pooling are proposed. However, their performance is far below the best part-based models [17], which limits this light-weight approximation to be further used in challenging recognition tasks.

In this paper, we propose a deep bilinear transformation (DBT) block, which can be integrated into a deep convolutional neural network (e.g., ResNet-50/101[18]), thus pairwise interactions can be learned in multiple layers to enhance feature discrimination ability. We empirically show that the proposed network is able to improve classification accuracy by calculating bilinear transformation over the most discriminative feature channels for a region while maintaining computational complexity.

Specifically, DBT consists of a semantic grouping layer and a group bilinear layer. First, the semantic grouping layer maps input channels into uniformly divided groups according to their semantic information (e.g., head, wings, and feet for a bird), thus the most discriminative representations for a specific semantic are concentrated within a group. Such representations can be further enhanced with pairwise interactions by the group bilinear layer, which conducts bilinear transformation within each semantic group. Since the bilinear transformation increases feature dimensions in each group, we obtain the output of group bilinear layer by conducting inter-group aggregation, which can significantly improve the efficientness of output channels. Group index encoding is obtained for preserving the information of group order during the aggregating process, and shortcut connection is adopted for residual learning of the original feature and bilinear feature. Compared to traditional bilinear transformation, DBT heavily relieves computational cost via grouping and aggregating, which ensures that it can be deeply integrated into CNNs. We summarize our contributions as follows:

- We propose deep bilinear transformation (DBT), which can be integrated into CNN blocks to obtain deep bilinear transformation network (DBTNet), thus pairwise interactions can be learned in multiple layers to enhance feature discrimination ability.

- We propose to obtain pairwise interaction only within the most discriminative feature channels for an image position/region by learning semantic groups and calculating intra-group bilinear transformation.

- We conduct extensive experiments to demonstrate the effectiveness of DBTNet, which can achieve new state-of-the-arts on three challenging fine-grained datasets, i.e., CUB-Bird, Stanford-Car, and FGVC-Aircraft.

The rest of the paper is organized as follows. We discuss the relation of our approach to recent works in Section 2, and we present our approach in Section 3. We then evaluate and report results in Section 4 and conclude the paper with Section 5.

## 2    Related Work

### 2.1    Fine-Grained Image Recognition

**Bilinear pooling.** Bilinear pooling [10] is proposed to obtain rich and orderless global representation for the last convolutional feature, which achieved the state-of-the-art results in many fine-grained datasets. However, the high-dimensionality issue is caused by calculating pairwise interaction between channels, thus dimension reduction methods are proposed. Specifically, low-rank bilinear pooling [15] proposed to reduce feature dimensions before conducting bilinear transformation, and compact bilinear pooling [14] proposed a sampling based approximation method, which can reduce feature dimensions by two orders of magnitude without performance drop. Different from them, we reduce feature dimension by intra-group bilinear transformation and inter-group aggregating, and a detailed discussion can be found in Section 3.4. Moreover, feature matrix normalization [11–13] (e.g., matrix square-root normalization) is proved to be important for bilinear feature, while we do not use such technics in our deep bilinear transformation since calculating such root is expensive and not practical to be deeply stacked in CNNs. Second-order pooling convolutional networks[19] also proposed to integrate bilinear interactions into convolutional blocks, while they only use such bilinear features for weighting convlutional channels.

**Weakly-supervised part learning.** Semantic parts play an important role in fine-grained image recognition, which adopts a divide and conquer strategy to learn fine-grained details in part level and conduct part alignment for recognition. Recent part learning methods use attention mechanism to learn semantic parts in a weakly-supervised manner. Specifically, TLAN [20] proposed to obtain part templates on both objects and parts by clustering CNN filters, DVAN [21] proposed to explicitly pursue the diversity of attention and is able to gather discriminative information to the maximal extent, and MA-CNN [9] proposed to learn multiple attentions for each image by grouping convolutional channels semantically, which is further optimized by a diversity loss and a distance loss. Inspired by such methods, we propose a simple yet effective semantic grouping constrain to integrate semantic information into bilinear features.

### 2.2    Group Convolution

Group convolution is first proposed in AlexNet [22], which can distribute the model over two GPUs to solve the problem of out of GPU memory. Such a grouping operation is rethought and further

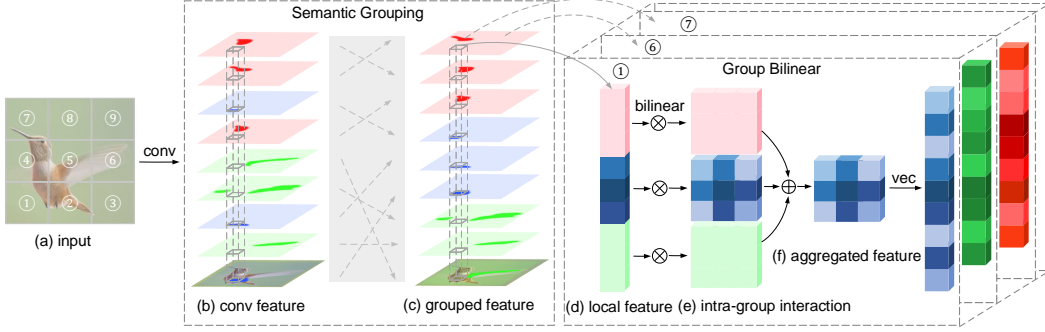

Figure 1: An overview of the proposed deep bilinear transformation. Given an input of image in (a), Semantic Grouping module learns to group relevant feature channels, according to their corresponding regions. For example in (b) and (c), pink and green channel corresponds to head and wings, respectively. Group Bilinear further calculates and aggregates intra-group pairwise interactions in (d), (e), and (f). We can observe that the bilinear feature of a part is obtained by the most discriminative feature channels for this part. [Best viewed in color]

studied in ResNeXt [23], which is an effective and efficient way to reduce convolutional parameters with even better performance. Such a method is widely used in efficient network designing, such as MobileNet [24] and ShuffleNet [25]. CondenseNet [26] takes a step further to propose a group learning strategy, instead of simply group convolutional channels by their index order. Compared with them, our novelty lies in two folds: 1) we integrate semantic information into the grouping process, and 2) we are the first to adopt channel grouping for bilinear transformation.

## 3    Deep Bilinear Transformation

Bilinear pooling is proposed to capitalize pairwise interactions among feature elements by outer product. We denote an input convolutional feature as $\mathbf{X} \in \mathbb{R}^{N \times HW}$, where $H$, $W$, and $N$ are the hight, width, and channel numbers, respectively. Thus the bilinear pooling with a fully connected layer can be denoted as:

$$\mathbf{f} = \mathbf{W}\frac{1}{HW}\text{vec}(\mathbf{X}\mathbf{X}^T) + \mathbf{b} = \mathbf{W}\frac{1}{HW}\sum_{i=1}^{HW}\text{vec}(\mathbf{x}_i\mathbf{x}_i^T) + \mathbf{b}, \tag{1}$$

where $\mathbf{W} \in \mathbb{R}^{K \times N^2}$, $\mathbf{b} \in \mathbb{R}^K$, and $\mathbf{f} \in \mathbb{R}^K$ are the weight, bias, and output of the fully connected layer, and $\mathbf{x}_i$ is the $i^{th}$ column of $\mathbf{X}$. Such a bilinear representation is proved to be powerful for many tasks [10, 27, 28]. However, the second order information is only obtained in the last convolutional layer, and the feature dimensionality larger than global average pooled [29] feature by $N$ times.

We integrate semantic information into bilinear features, and the proposed deep bilinear transformation (DBT) can be stacked with convolutional layers. For example, the concrete formulation with a $1 \times 1$ convolutional layer is given by:

$$\mathbf{f} = \mathbf{W}[\mathbf{y}_1, \mathbf{y}_2, ..., \mathbf{y}_{HW}] + \mathbf{b}, \text{ where}$$
$$\mathbf{y}_i = \mathcal{T}_B(\mathbf{A}\mathbf{x}_i) = \text{vec}(\sum_{j=1}^{G}((\mathbf{I}_j\mathbf{A}\mathbf{x}_i + \mathbf{p}_j)(\mathbf{I}_j\mathbf{A}\mathbf{x}_i + \mathbf{p}_j)^T)). \tag{2}$$

In the above equation, $\mathcal{T}_B(\cdot)$ is group bilinear function, $\mathbf{A}\mathbf{x}_i$ is semantically grouped feature, $G$ is the number of group, $\mathbf{p}_j$ is group index encoding vector, which indicates the group order, $\mathbf{I}_j \in \mathbb{R}^{\frac{N}{G} \times N}$ is a block matrix with $G$ blocks, whose $j^{th}$ block is an identity matrix $\mathbf{I}$ while others are zero matrixes, and $\mathbf{A}$ is semantic mapping matrix, which groups channels representing the same semantic together. Note that both the dimension of parameters $\mathbf{W} \in \mathbb{R}^{K \times (\frac{N}{G})^2}$ and the features $[\mathbf{y}_1, \mathbf{y}_2, ..., \mathbf{y}_{HW}] \in \mathbb{R}^{(\frac{N}{G})^2 \times HW}$ are reduced by $G^2$ times. We will introduce the detail for each item in this section.

## 3.1 Semantic Grouping Layer

It has been observed in previous work [7, 9, 20, 30, 31] that convolutional channels in high-level layers tend to have responses to specific semantic patterns. Thus we can divide convolutional channels into several groups by their semantic information. Specifically, given a convolutional feature $\mathbf{X} \in \mathbb{R}^{N \times HW}$, we denote each channel as a feature map $\mathbf{m}_i \in \mathbb{R}^{HW}$, where $i \in [1, N]$. We divide the semantic space into $G$ groups, and $\mathcal{S}(\mathbf{m}_i) \in [1, G]$ is a mapping function that maps a channel into a semantic group. The convolutional channels are uniformly grouped, i.e., $\mathcal{S}(\mathbf{m}_i) = \mathcal{S}(\mathbf{m}_j)$ if $\lfloor i/G \rfloor = \lfloor j/G \rfloor$. To obtain bilinear feature for semantic parts, we first arrange the channels in the order of semantic groups by:

$$[\tilde{\mathbf{m}}_1, \tilde{\mathbf{m}}_2, ..., \tilde{\mathbf{m}}_N] = [\mathbf{m}_1, \mathbf{m}_2, ..., \mathbf{m}_N]\mathbf{A}^T, \quad s.t. \quad \mathcal{S}(\tilde{\mathbf{m}}_i) = \lfloor i/G \rfloor, \tag{3}$$

where $\mathbf{A}^T \in \mathbb{R}^{N \times N}$ is a semantic mapping matrix, which needs to be optimized.

Since different semantic parts are located in different regions of a given image, which correspond to different positions in an convolutional feature, we can use such spacial information for semantic grouping. Specifically, the channels in the same/different semantic groups are optimized to share large/small overlaps of the response in spacial, which is formulated as semantic grouping loss $L_g$:

$$L_g = L_{intra} + L_{inter} = \sum_{\substack{0 \leq i,j < N \\ \lfloor i/G \rfloor = \lfloor j/G \rfloor}} -d_{ij}^2 + \sum_{\substack{0 \leq i,j < N \\ \lfloor i/G \rfloor \neq \lfloor j/G \rfloor}} d_{ij}^2, \tag{4}$$

where the pairwise correlation is $d_{ij} = \frac{\tilde{\mathbf{m}}_i^T \tilde{\mathbf{m}}_j}{\|\tilde{\mathbf{m}}_i\|_2 \cdot \|\tilde{\mathbf{m}}_j\|_2}$.

Note that such a semantic grouping can be implemented by a $1 \times 1$ convolutional layer. Specifically, a convolutional layer can be denoted as $\mathbf{x} = \mathbf{W}\mathbf{z}$, where $\mathbf{z} \in \mathbb{R}^M$ is the input feature, $\mathbf{x} \in \mathbb{R}^N$ is the output feature, and $\mathbf{W} \in \mathbb{R}^{N \times M}$ is the weight matrix. Let $\mathbf{U} = \mathbf{A}\mathbf{W}$, the mapped feature can be obtained by $\mathbf{A}\mathbf{x} = \mathbf{A}\mathbf{W}\mathbf{z} = \mathbf{U}\mathbf{z}$, thus $\mathbf{U}$ is the weight matrix of a semantic grouping layer, whose outputs are semantically grouped. Note that $\mathbf{U}$ is used for not only semantic grouping, but also convolutional feature learning, thus we can uniformly divide the output channels into a preset number of groups and obtain the grouped features in the CNN training process.

## 3.2 Group Bilinear Layer

Given a semantically grouped convolutional feature, we propose a group bilinear layer to efficiently generate bilinear features without increasing feature dimensions. Specifically, group bilinear layer calculates bilinear transformation over the channels within each group, which can enhance the representation of the corresponding semantic by pairwise interactions. Note that the intra-group bilinear transformation increases feature dimensions in each group, to improve the efficientness of output channels, we further aggregate such intra-group bilinear features. Thus the proposed group bilinear can be obtained as:

$$\mathbf{y} = \mathcal{T}_B(\mathbf{A}\mathbf{x}) = \text{vec}(\sum_{j=1}^{G}((\mathbf{I}_j\mathbf{A}\mathbf{x})(\mathbf{I}_j\mathbf{A}\mathbf{x})^T)), \tag{5}$$

where $\mathbf{A}$ is the mapping matrix learned in Equation 3, and $\mathbf{I}_j \in \mathbb{R}^{\frac{N}{G} \times N}$ is a block matrix with $G$ blocks, whose $j^{th}$ block is an identity matrix $\mathbf{I}$ while others are zero matrixes. $\mathbf{I}_j\mathbf{A}\mathbf{x}$ is the $j^{th}$ group of the semantically grouped feature. The dimension of the input feature $\mathbf{x} \in \mathbb{R}^N$ is $N$, and that of the output feature $\mathbf{y} \in \mathbb{R}^{(\frac{N}{G})^2}$ is $(\frac{N}{G})^2$, we adopt $G = \sqrt{N}$ (conduct bilinear interpolation over channels for non-integer cases) to keep feature dimensions unchanged, thus DBT can be conveniently integrated into CNNs without changing original network architecture.

Conducting aggregating over groups can reduce feature dimensionality by $G$ times, while the information of group order would be lost in such a process. Thus we introduce position encoding [32] into convolutional channel representations, and add a group index encoding item to preserve group order information and improve discriminations for the features in different groups:

$$\begin{aligned} \mathbf{P}_j(2i) &= \sin(j/t^{2i/\frac{N}{G}}), \\ \mathbf{P}_j(2i+1) &= \cos(j/t^{2i/\frac{N}{G}}), \end{aligned} \tag{6}$$

where $t$ indicates the frequency of function $\sin(\cdot)$. Such a group index encoding is element-wise added into the $j^{th}$ group feature before conducting bilinear transformation: $(\mathbf{I}_j\mathbf{A}\mathbf{x} + \mathbf{P}_j)(\mathbf{I}_j\mathbf{A}\mathbf{x} + \mathbf{P}_j)^T$, thus the item of $\mathbf{P}_j(\mathbf{I}_j\mathbf{A}\mathbf{x})^T$ can preserve the group index information in the aggregating process. To this end, the proposed semantic group bilinear module can be obtained, which is shown in Equation 2.

### 3.3 Deep Bilinear Transformation Network

**Activation and shortcut connection.** Non-linear activation functions can increase the representative capacity of a model. Instead of $ReLU$, $tanh$ function is widely used to activate bilinear features, because such functions are able to suppress large second-order values. Moreover, inspired by residual learning [18], we add shortcut connections for semantic group bilinear feature to assist optimization:

$$\mathbf{f} = \mathbf{W}(BN(tanh(\mathcal{T}_B(\mathbf{A}\mathbf{X})) + \mathbf{X}) + \mathbf{b}. \tag{7}$$

Such a shortcut connection 1) fuse original and bilinear feature, and 2) build a gateway for the original feature for backward propagation. Note that we make the network study from the original feature by initializing the "scale" parameter of the batch normalization layer to be zero, which is an effective way for parameters optimization.

**Deep bilinear transformation block.** Figure 2 shows the network structure of deep bilinear transformation block. The semantic grouping layer is a $1 \times 1$ convolution with the constraints introduced in Equation 4. $3 \times 3$ convolutional layers are the key components for feature extraction in CNNs, which can integrate local context information into convolutional channels. The group bilinear layer is followed by a $3 \times 3$ convolution, thus the rich pairwise interactions can be further integrated to obtain fine-grained representation.

**Network architecture.** The proposed semantic group bilinear module can be integrated into deep convolutional neural networks. Table 1 is an illustration of integrating DBT into each ResNet block, and the effectiveness of DBT in different stages are discussed in Section 4.2. The overall loss $L$ for such a model is shown as:

$$L = L_c + \lambda \sum_b^B L_g^{(b)}, \tag{8}$$

where $L_c$ is softmax cross entropy loss for classification, $L_g^{(b)}$ is semantic grouping loss over the $b^{th}$ block, $B$ is the number of residual blocks, and $\lambda$ is the weight of semantic grouping loss.

Since our DBT does not change feature dimensions (as introduced in Section 3.2), it can be conveniently integrated into convolutional blocks. We conduct DBT before $3 \times 3$ convolutional layers and the value of $G$ indicates the number of semantic groups. Note that the proposed DBT is efficient since there are no additional parameters, and the computational cost is also very low, i.e., 5M FLOPs.

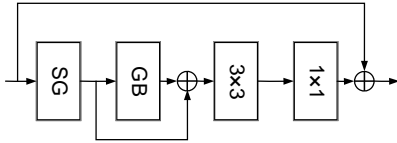

Figure 2: An illustration of the structure of deep bilinear transformation block, where "SG" indicates semantic grouping layer, "GB" indicates group bilinear layer, $1 \times 1$ and $3 \times 3$ indicates the kernel size of convolutional layers.

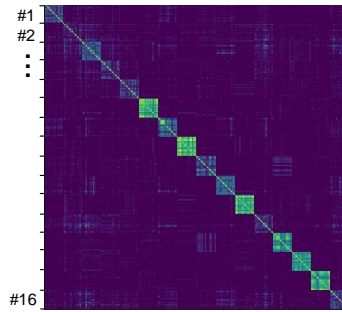

Figure 3: An illustration of intra-group (i.e., diagonal blocks in the figure) and inter-group pairwise interaction. Note that yellow indicates large value and purple indicates small value.

### 3.4 Discussion

The proposed deep bilinear transformation can generate fine-grained representation without increasing feature dimension by calculating intra-group bilinear transformation and conducting inter-group aggregation. In this subsection, we analyze the intra-group and inter-group pairwise interaction and show the difference between our work and previous low-dimensional bilinear variants.

Table 1: An illustration of integrating DBT into deep CNNs. "SG" indicates semantic grouping layer, and "GB" indicates group bilinear layer. "$G = 16$" suggests channels are divided into 16 semantic groups. A detailed discussion on the integrating stages can be found in Section 4.2.

| Stage | Output | ResNet-50 [18] | DBTNet-50 | DBTNet-101 |
|---|---|---|---|---|
| I | $112 \times 112$ | \multicolumn{3}{c}{$7 \times 7, 64$, stride 2} | | |
| II | $56 \times 56$ | \multicolumn{3}{c}{$3 \times 3$ max pool, stride 2} | | |
| II | $56 \times 56$ | $\begin{bmatrix} 1 \times 1, 64 \\ 3 \times 3, 64 \\ 1 \times 1, 256 \end{bmatrix} \times 3$ | $\begin{bmatrix} SG, 1 \times 1, 64 \\ GB, 64, G = 8 \\ 3 \times 3, 64 \\ 1 \times 1, 256 \end{bmatrix} \times 3$ | $\begin{bmatrix} SG, 1 \times 1, 64 \\ GB, 64, G = 8 \\ 3 \times 3, 64 \\ 1 \times 1, 256 \end{bmatrix} \times 3$ |
| III | $28 \times 28$ | $\begin{bmatrix} 1 \times 1, 128 \\ 3 \times 3, 128 \\ 1 \times 1, 512 \end{bmatrix} \times 4$ | $\begin{bmatrix} SG, 1 \times 1, 128 \\ GB, 128, G = 8 \\ 3 \times 3, 128 \\ 1 \times 1, 512 \end{bmatrix} \times 4$ | $\begin{bmatrix} SG, 1 \times 1, 128 \\ GB, 128, G = 8 \\ 3 \times 3, 128 \\ 1 \times 1, 512 \end{bmatrix} \times 4$ |
| IV | $14 \times 14$ | $\begin{bmatrix} 1 \times 1, 256 \\ 3 \times 3, 256 \\ 1 \times 1, 1024 \end{bmatrix} \times 6$ | $\begin{bmatrix} SG, 1 \times 1, 256 \\ GB, 256, G = 16 \\ 3 \times 3, 256 \\ 1 \times 1, 1024 \end{bmatrix} \times 6$ | $\begin{bmatrix} SG, 1 \times 1, 256 \\ GB, 256, G = 16 \\ 3 \times 3, 256 \\ 1 \times 1, 1024 \end{bmatrix} \times 23$ |
| V | $7 \times 7$ | $\begin{bmatrix} 1 \times 1, 512 \\ 3 \times 3, 512 \\ 1 \times 1, 2048 \end{bmatrix} \times 3$ | $\begin{bmatrix} SG, 1 \times 1, 512 \\ GB, 512, G = 16 \\ 3 \times 3, 512 \\ 1 \times 1, 2048 \end{bmatrix} \times 3$ | $\begin{bmatrix} SG, 1 \times 1, 512 \\ GB, 512, G = 16 \\ 3 \times 3, 512 \\ 1 \times 1, 2048 \end{bmatrix} \times 3$ |
| | $1 \times 1$ | \multicolumn{3}{c}{global average pool, 1000-d fc, softmax} | | |
| (#Params, FLOPs) | | (25.5 M, 3.8 G) | (25.5 M, 3.8 G) | (44.4 M, 7.6 G) |

**Intra-group and inter-group pairwise interaction** We empirically show the intra-group and inter-group pairwise interaction in Figure 3. Specifically, we extract semantically grouped convolutional features in stage3 of DBTNet-50 for all the testing samples in CUB-200-2011 and visualize the average pairwise interaction in Figure 3. There are 256 channels in 16 groups, and yellow indicates large value while purple indicates small value. It can be observed that intra-group interaction plays a dominant role. The small intra-group interactions (e.g., group #2) indicate that such semantic parts appear less than others. Consider two convolutional channels from different semantic groups, i.e., $\mathbf{m}_i, \mathbf{m}_j \in \mathbb{R}^{HW}$, and $\mathcal{S}(\mathbf{m}_i) \neq \mathcal{S}(\mathbf{m}_i)$. Since different semantic parts are located in different positions, we can obtain that $\mathbf{m}_i \circ \mathbf{m}_j = \mathbf{0}$, where $\circ$ indicates element-wise product, and $\mathbf{0} \in \mathbb{R}^{HW}$. In a word, the bilinear feature among channels in different semantic groups is zero vector, which enlarges the bilinear feature dimension without providing discriminative information. Our proposed DBT solve this problem by conducting outer product over channels within a group. However, previous bilinear variants with low dimensions cannot solve this problem.

**Compared with low-dimensional bilinear variants** Low-rank bilinear [15] proposes to reduce feature channels by a $1 \times 1$ convolutional layer before conducting bilinear pooling, however, the reduced channels are still in different semantic groups. Compact bilinear [14] proposes to use sampling methods to approximate the second-order kernel. The random maclaurin projection version of compact bilinear pooling can be denoted as: $\mathbf{f} = \mathbf{W}_1 \mathbf{x} \circ \mathbf{W}_2 \mathbf{x}$, where $\circ$ indicates hadamard product, and $\mathbf{W}_1, \mathbf{W}_2$ are random and fixed, whose entries are either $+1$ or $-1$ with equal probability. Hadamard low-rank bilinear [27] takes one step further to use learnable weights: $\mathbf{f} = \mathbf{P}^T(\mathbf{U}\mathbf{x} \circ \mathbf{V}\mathbf{x}) + \mathbf{b}$, where $\mathbf{P}, \mathbf{U}, \mathbf{V}$, and $\mathbf{b}$ are learnable parameters. We take one element of $\mathbf{y} = \mathbf{U}\mathbf{x} \circ \mathbf{V}\mathbf{x}$ for analysis, which is denoted as $y = \mathbf{u}^T \mathbf{x} \circ \mathbf{v}^T \mathbf{x} = \sum_{j,k} u_j v_k (x_j x_k)$. Compared to original bilinear $y_i = \mathbf{x}^T \mathbf{W} \mathbf{x} = \sum_{j,k} W_{j,k} (x_j x_k)$, hadamard low-rank bilinear decompose parameter matrix $\mathbf{W}$ into two vectors $\mathbf{u}, \mathbf{v}$. Such approximate also contains uninformative zeros as $x_j x_k = 0$ if the $j^{th}$ channel and the $k^{th}$ channel are not in the same semantic group.

## 4 Experiments

### 4.1 Experiment setup

**Datasets:** We conducted experiments on three widely used fine-grained datasets (i.e., CUB-200-2011 [2] with 6k training images for 200 categories, Stanford-Car [3] with 8k training images for 196 categories and FGVC-Aircraft[33] with 6k training images for 100 categories) and a large scale fine-grained dataset iNatualist-2017 [34] with 600k training images for 5,089 categories. Note that

traditional fine-grained task focus on distinguishing categories within a super-category, while there are 13 super categories in iNaturalist.

**Implementation:** We use MXNet [35] as our code-base, and all the models are trained on 8 Tesla V-100 GPUs. We follow the most common setting in fine-grained tasks to pre-train the models on ImageNet [36] with input size of $224 \times 224$, and fine-tune on fine-grained datasets with input size of $448 \times 448$ (unless specially stated). We adopt consine learning rate schedule, SGD optimizer with the batch size to be 48 per GPU. The weight for semantic constrain in Equation 4 is set to 3e-4 in pre-training stage and 1e-5 in fine-tune stage. And we defer other implementation details to our code `https://github.com/researchmm/DBTNet`.

### 4.2 Ablation studies

We conduct ablation studies for DBTNet-50 on CUB-200-2011, with input image size of $224 \times 224$.

**Semantic grouping** The proposed semantic grouping constrain in Equation 4 encourages the channels with similar semantic to gather together, which is a vital pre-processing for group bilinear. The impact of the loss weight (i.e., $\lambda$ in Equation 8) is shown in Table 2 for both pre-training stage and fine-tuning stage. It can be observed that such a constraint has a significant impact in the pre-training stage. Specifically, without semantic grouping constraint, that is, conduct group bilinear over randomly grouped channels, bring a $4.8\%$ accuracy drop compared to with suitable constraint. While a large constraint would also damage classification accuracy because the classification loss is supposed to dominate the optimization of the network. A similar phenomenon can be observed in the fine-tuning stage, while the impact is much less, which indicates that the semantic grouping is important in the early stage of network optimization.

| loss weight | accuracy (%) |
|---|---|
| (0,0) | 79.8 |
| (3e-4,0) | 84.6 |
| (3e-3,0) | 83.1 |
| (3e-4,1e-5) | **85.1** |
| (3e-4,1e-4) | 84.8 |

Table 2: Ablation study on semantic grouping constraint. In the form of (pre-train, fine-tune).

| group index | t | accuracy (%) |
|---|---|---|
| w/o | - | 84.5 |
| w/ | 1.1 | 85.0 |
| | 1.5 | **85.1** |
| | 2 | 84.8 |
| | 10 | 84.5 |

Table 3: Ablation study on group index encoding. $t$ indicates the frequency in Equation 6.

| stage | shortcut | accuracy (%) |
|---|---|---|
| V | w/o | 84.6 |
| V | w/ | 84.8 |
| V + IV | w/o | 83.1 |
| V + IV | w/ | **85.1** |

Table 4: Ablation study on shortcut connection. Note that all settings include 'last layer' mensioned in Table 5.

| approach | accuracy (%) |
|---|---|
| baseline | 83.3 |
| last layer | 84.4 |
| Stage V | 84.5 |
| last layer + Stage V | 84.8 |
| last layer + Stage V + IV | **85.1** |
| last layer + Stage V + IV + III | 85.0 |

Table 5: Ablation study on integrated stages.

| grouping mechanism | w/o constraints | constraints in MA-CNN [9] | ours (Equation 4) |
|---|---|---|---|
| accuracy (%) | 79.8 | 83.2 | **85.1** |

Table 6: Comparison on channel grouping constrains.

**Group index encoding** Group index encoding is obtained before conducting aggregation over groups, which can preserve group index information in the aggregated feature. Table 3 shows the impact of group index encoding with different frequencies (i.e., $t$ in Equation 6). It can be observed that group index encoding can improve classification by $0.6\%$ accuracy gains, and the frequency $t$ should be small since the encoding dimension, i.e., $\frac{N}{G}$, is small (typically 16 or 32).

**Shortcut connection** Shortcut connection can facilitate the network from two aspects, i.e., 1) fusing original and bilinear features, and 2) enabling a straight backward propagation. Table 4 shows the impact of shortcut connection for semantic group bilinear block in different stages. It can be observed that shortcut connection can bring $0.2\%$ accuracy gains in stage 4, and $2.0\%$ accuracy gains in stage $3 + 4$. And note that without shortcut connection, adding semantic group bilinear in stage 3 brings accuracy drop. Such a drop is caused by optimization problem, which can be solved by the shortcut

Table 7: Comparison in terms of classification accuracy on the CUB-200-2011, Stanford-Car, and FGVC-Aircraft datasets. The "dimension" indicates the dimension of the last layer bilinear feature.

|  | approach | dimension | CUB-200-2011 | Stanford-Car | Aircraft |
|---|---|---|---|---|---|
| ResNet-50 | Compact Bilinear [14] | 14k | 81.6 | 88.6 | 81.6 |
|  | Kernel Pooling [37] | 14k | 84.7 | 91.1 | 85.7 |
|  | iSQRT-COV [13] | 8k | 87.3 | 91.7 | 89.5 |
|  | iSQRT-COV [13] | 32k | 88.1 | 92.8 | 90.0 |
|  | DBTNet-50 (ours) | 2k | 87.5 | **94.1** | **91.2** |
| ResNet-101 | DBTNet-101 (ours) | 2k | **88.1** | **94.5** | **91.6** |

connection together with batch normalization, where the "scale" parameter is supposed to be initialized with zero.

**Integrated stages** Table 5 shows the ablation study on integrated stages. It can be observed that deeply integrating DBT into Stage V and Stage IV brings $0.7\%$ accuracy gains compared to only conducting it over the last layer. Since the proposed DBT taking advantages of semantic information, integrating DBT into Stage III with low-level features can not further improve the performance. Thus we integrate DBT into Stage IV, Stage V, together with the last layer in our DBTNet.

**Grouping constrains** The proposed group bilinear requires the intra-group channels to be highly correlated, and the proposed semantic grouping can better satisfy such requirements than MA-CNN [9]. Specifically, MA-CNN [9] adopts the idea of k-means, which optimizes each channel to its cluster center. While the proposed grouping method in this paper optimize the correlation of intra-group and inter-group channels in a pairwise manner (as shown in Equation 4). Moreover, we conducted experiments by replacing our grouping loss with MA-CNN [9], and the results in Table 6 also show the effectiveness of our proposed grouping module.

### 4.3 Comparison with the state-of-the-art

**Fine-grained image recognition benchmarks**: Table 7 shows the comparison with other bilinear based fine-grained recognition methods on three competitive datasets, i.e., CUB-200-2011 [2], Stanford-Car [3] and FGVC-Aircraft[33], with input image size of $448 \times 448$. Since our DBT block is basically built on ResNet block, we compared with other methods which also uses ResNet as the backbone. It can be observed that the proposed DBT can significantly outperform Compact bilinear [14] and Kernel Pooling [37] in all three datasets. Compared to the state-of-the-art iSQRT-COV [13], which conducts matrix normalization over bilinear features, we can also obtain a large margin of accuracy gains on two of the three datasets, i.e., Stanford-Car and FGVC-Aircraft. Moreover, we can also see better performance by integrating DBT into deeper ResNet-101. For DBTNet-101, we integrated DBT into the last layer, the Stage V and the last 6 layers of Stage IV.

**Large-scale image recognition benchmarks**: To further evaluate the proposed DBTNet on large scale image datasets, we conduct experiments on iNaturalist-2017 [34]. We compare the performance of ResNet-50 and DBTNet-50 with $224 \times 224$ input images, and observed that DBTNet-50 ($62.0\%$) can outperform ResNet-50 ($59.9\%$) with $2.1\%$ accuracy gains. Moreover, our DBTNet-50 can obtain $1.6\%$ accuracy gains over ResNet-50 on ImageNet [36] dataset for general image recognition tasks.

## 5 Conclusion

In this paper, we propose a novel deep bilinear transformation (DBT) block, which can be integrated into deep convolutional neural networks. The DBT takes advantages of semantic information and can obtain bilinear features efficiently by calculating pairwise interaction within a semantic group. A highly-modularized network DBTNet can be obtained by stacking DBT blocks with convolutional layers, and the deeply integrated bilinear representations enable DBTNet to achieve new state-of-the-art in several fine-grained image recognition tasks. Since semantic information can only be obtained in high-level features, we will study on conducting deep bilinear transformation over low-level features in the future. Moreover, we will explore to integrate matrix normalization into DBT in an efficient way, to further leverage the bilinear representations.

**Acknowledgement:** This work was supported by the National Key R&D Program of China under Grant 2017YFB1300201, the National Natural Science Foundation of China (NSFC) under Grants 61622211 and 61620106009 as well as the Fundamental Research Funds for the Central Universities under Grant WK2100100030.

## Footnotes

*This work was performed when Heliang Zheng was visiting Microsoft Research as a research intern.

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
