[Reviews · NeurIPS 2019]

Reviewer 1



Updates: I appreciate the authors' effort to provide additional experiments. Now I understand the difference and significance of the proposed grouping method compared to MA-CNN, as well as its effectiveness on large-scale ImageNet dataset. On the other hand, I still feel the core methodological contribution is marginal because pairwise clustering is an well-studied approach to clustering in general, and I think switching to it is not a surprising direction. Overall, I would like to set my final evaluation as borderline negative. This paper presents a method combining bilinear pooling and channel grouping for fine-grained image recognition. Overall, the idea itself is quite reasonable as it can significantly reduce the dimension of resultant features, and the results also seem promising. However, because the idea and effectiveness of channel grouping for fine-grained recognition are proposed in MA-CNN [9] and not novel, the contribution of the paper seems rather incremental: just performs the standard bilinear pooling on top of the grouped features. Moreover, the presentation of the paper is not very clear and I cannot see some crucial points as I commented in "5.Improvements". For those reasons, my initial impression is leaning toward rejection.

Reviewer 2



This paper proposes a bilinear transformation module which combines semantic group of feature channels and intra-group bilinear pooling. This module does not increase the number of channels while improves non-trivially the network performance. As far as I know, the proposed bilinear transformation is clearly different from existing works. The semantic grouping layer is important in the proposed bilinear transformation module. How is the constraint as described in Eq. (4) is implemented as a loss? Will such a constraint will be imposed and implemented in the final loss for every residual block (i.e. bottleneck block)? Note that in ResNet-50 (resp. 101) has 16 (resp. 33) residual blocks. I suggest the authors make these clear and write out explicitly the formula of the whole loss. The experiments are basically extensive and the results are competitive. Nevertheless, I have some questions which hopefully could be addressed by the authors. In Table 5 stage 3+4 (w/ shortcut) produces an accuracy of 85.1, while in Table 6, last layer+stage V+stage IV produces the same accuracy. Is the accuracy of 85.1 obtained by combination of exactly what layers? What does last layer in Table 6 indicate? Are the proposed models pre-trained on large-scale ImageNet? If so, I would like to know the comparison of recognition accuracy on ImageNet with vanilla ResNet and other second-order pooling methods. I would also like to know the data augmentation method used. Is the random crop used in training? The proposed bilinear transform can be inserted into lower layers of deep networks, different from the classical the bilinear modules which can only be inserted at the end. In this aspect, there are some related works missing in the paper, e.g., Factorized Bilinear Models in ICCV 17 and global second-order convolutional networks in CVPR 19. ------------------------------------------ In the rebuttal, the authors have given explicitly the proposed loss function, making clear the mechanism of semantic grouping as well as its difference from the grouping method of MA-CNN. The performance on ImageNet is very impressive, showing that the proposed method is not only suitable for fine-grained classification but also general, large-scale visual recognition. I suggest the authors add the modifications into the manuscript; some typos or grammatical errors should also be corrected (e.g., line 76, aliment-->alignment; line 221, which a vital-->which is a vital). In summary, the paper proposed a compact bilinear pooling layer that can be inserted into throughout a network, clearly different from previous bilinear pooling methods. The performance is also very competitive, on both fine-grained classification and large-scale visual recognition.

Reviewer 3



Clarity. - The paper is well-written and the method is well-explained. Originality. - The proposed extension is very well-motivated, but the paper itself is incremental. As the major contribution seems from limiting pair-wise interactions within each semantic group. Significance - The proposed extension to bilinear transformation is well-motivated, and, to certain extent, combines advantages of bilinear pooling and part-based learning. - The proposed method looks a good practical approach for very good fine-grained classification results. =========================== Thank authors for the response! I have read the author response and my opinion remains the same, as I still feel the contribution seems a bit incremental over existing work.

[Author Response · NeurIPS 2019]

Thanks for the reviewers' valuable comments. We appreciate the positive comments on well-motivated approach with
promising performance for fine-grained image recognition. Moreover, we can observe improvements on large scale
*ImageNet* recognition task (as shown in the table for Reviewer #2). We address the concerns of reviewers as following.

**To Reviewer #1:**

**Q 1.1 Compared with MA-CNN [9].** The proposed group bilinear requires the intra-group channels to be highly
correlated (refer to the definition in Q 3.1), and the proposed semantic grouping can better satisfy such requirements
than MA-CNN [9]. Specifically, [9] adopts the idea of k-means, which optimizes each channel to its cluster center.
While the proposed grouping method in this paper optimize the correlation of intra-group and inter-group channels
in a **pairwise manner** (as shown in Q 1.3 $L_g$), which has been proved to be able to obtain a tighter cluster (higher
correlations), e.g., *mixture modelling by affinity propagation* Brendan Frey et al., nips 2006 and *clustering by passing*
*messages between data points*, Brendan Frey et al., science 2007.
Moreover, we conducted experiments by replacing our grouping loss with [9], and the results also show the effectiveness
of our proposed grouping module (i.e., one of the main contributions of this paper):

| Grouping Mechanism | Grouping w/o constraints | Constraints in MA-CNN [9] | Constraints in DBTNet (ours) |
|---|---|---|---|
| Accuracy (%) | 79.8 | 83.2 | **85.1** |

**Q 1.2 The concrete loss function.** Eq. (4) is exact the criterion to optimize the parameter $\mathbf{A}$, and it can be formulated
as: $L_g = L_{intra} + L_{inter}$, where $L_{intra} = \sum\limits_{\substack{0 \le i,j < N \\ \lfloor i/G \rfloor = \lfloor j/G \rfloor}} -d_{ij}^2$ and $L_{inter} = \sum\limits_{\substack{0 \le i,j < N \\ \lfloor i/G \rfloor \ne \lfloor j/G \rfloor}} d_{ij}^2$ are designed to maxi-
mize/minimize the intra/inter-group correlations, respectively. Note that the notations above are the same with Eqn.
(3), and the pairwise correlation is $d_{ij} = \frac{\tilde{\mathbf{m}}_i^T \tilde{\mathbf{m}}_j}{\|\tilde{\mathbf{m}}_i\|_2 \cdot \|\tilde{\mathbf{m}}_j\|_2}$. The overall loss $L$ is shown as: $L = L_c + \lambda \sum\limits_{b}^{B} L_g^{(b)}$, where $L_c$
is softmax cross entropy loss for classification, $L_g^{(b)}$ is semantic grouping loss over the $b^{th}$ block, $B$ is the number of
residual blocks, and $\lambda$ is the weight of semantic grouping loss. We will add these equations in the method section.

**Q 1.3 Constrains of the index mapping matrix.** Thanks for your comments. $\mathbf{A}$ is an approximate index mapping
matrix, whose rows are constrained to be (approximate) one-hot vectors via a $softmax$ with small "temperature".
For example, a vector $\mathbf{x}$ can be approximately transformed into a one-hot vector by: $softmax(\mathbf{x}/T)$, where $T$ is the
temperature and is set to 0.0001 in our experiments. We will add this missing detail in the method section.

**Q 1.4 Experiment settings for Table 7**. As described in Page 6, Line 214 and Line 219, we conduct ablation studies
with $224 \times 224$ input images for fast training and use $448 \times 448$ input images in Table 7 for fair comparison.

**To Reviewer #2:**

**Q 2.1 Loss function.** Thanks for your advice, and the concrete loss function can be found in Q 1.1 for Reviewer #1.

**Q 2.2 Inconsistent notations.** Thanks for your comments, and we will correct the notation "stage 3,4" into "Stage
IV,V" respectively. "Last layer" indicates conducting group bilinear over the last layer of the backbone, which is added
by default in Table 5. Thus "stage 3+4" in Table 5 is exact "last layer+Stage IV+Stage V" in Table 6.

**Q 2.3 Results on ImageNet.** The proposed models are pre-trained on ImageNet-1K. It can be observed that DBTNet-
50 outperforms Resnet-50 and iSQRT-COV-8k with an obvious margin (1.6% and 0.5% absolute improvements
respectively), and it achieves comparable results with iSQRT-COV-32k, whose feature dimension is 16 times larger:

| Approach | ResNet-50 [17] | iSQRT-COV [13] | iSQRT-COV [13] | DBTNet-50 (ours) |
|---|---|---|---|---|
| Dimension | 2k | 8k | 32k | 2k |
| Top-1 err. (the lower, the better) | 23.9 | 22.8 | 22.1 | **22.3** |

We use standard data augmentation methods provided by MXNet, i.e., random resized crop and random mirror.

**Q 2.4 Missing references.** Thanks for your advice, and we will add discussions for the missing references.

**To Reviewer #7:**

**Q 3.1 Definition of semantic groups**. A semantic group indicates a series of channels which represent the same
semantic pattern, that is, the channels within a semantic group have responses in the same positions for a given image.
Specifically, we obtain semantic groups by equally dividing 512 arranged channels (Eqn. (3)) into 16 groups and
optimizing the responses of intra/inter-group channels to share larger/smaller spacial overlaps by Eqn. (4).

**Q 3.2 Clarification for contributions**. The proposed group bilinear makes deep bilinear transformation **doable** and
the proposed semantic grouping ensures **competitive performance**. Designing suitable grouping methods plays a
key role. As shown in the table for Reviewer #1, different grouping mechanisms achieve different results with large
variances (79.8, 83.2, and 85.1). Specifically, the proposed semantic groups can enhance intra-group correlation, thus
rich pairwise interactions can be obtained by the intra-group bilinear; inter-group correlation is suppressed, which
makes the aggregation among groups free from information merging.
Moreover, such a design can also achieve promising performance on ImageNet task (see the table for Reviewer #2).

[Meta-Review · NeurIPS 2019]

The paper received mixed reviews because reviewers were confused about the method and novelty. After the rebuttal, two reviewers better understood and appreciated the results, as well as difference to the state-of-the-art. Two reviewers argue the novelty is limited. While the novelty is minor, there is novelty, which is articulated by the rebuttal and appreciated by the most favorable reviewer. The paper is accepted on the condition the information from the rebuttal is integrated into the camera ready.